# Assessment of healthcare financing by older adults during routine consultations in Cameroon

**Marie-José Essi**[1], **Johanne A. Abossolo**[1], **Divine H. Aba'a**[1], **Sidonie L. Ndjebet**[1], **Gabrièle Laborde-Balen**[2], **Bernard Taverne**[2], **Laura Ciaffi**[2]*

**1** Social Medicine Research Unity of Hepatitis and Health Communication Laboratory Research, Faculty of Medicine and Biomedical Sciences, University of Yaounde I, Cameroon, **2** TransVIHMI (Univ. de Montpellier, INSERM, IRD), Montpellier, France

* laura.ciaffi@ird.fr

## Abstract

Although older adults' care extends beyond strictly medical concerns, access to healthcare services remains a major challenge at this stage of life. The increasing demand for healthcare often coincides with declining functional abilities, cessation of income-generating activities, and insufficient coverage by social protection systems. This study aimed to estimate the medical expenses incurred during routine consultations by older adults in Yaoundé, in the absence of universal health coverage, and to identify the methods used to finance these healthcare costs. A descriptive, cross-sectional, quantitative study was conducted in four health facilities in Yaounde between February and March 2023. The study included individuals aged 60 and above who attended outpatient consultations at the selected facilities and were recruited after their medical appointments. A total of 170 participants were included, with a mean age of 68.4 years. Most respondents lived in socially and economically vulnerable conditions, particularly women. Cardiovascular, musculoskeletal, and urological conditions accounted for 70.5% of consultation reasons. Hypertension was the most common comorbidity (28.2%), followed by rheumatic disorders (21.1%) and diabetes (19.7%). The average cost of a routine medical consultation, including medications and diagnostic tests, was 42,958 XAF (approximately €65.49). In 90.6% of cases, care was financed through direct out-of-pocket household payments. In Cameroon, healthcare expenses for older adults are primarily borne by households, which limits access to care for those with low incomes. Implementing a more inclusive and equitable health system, integrated into a comprehensive social protection policy, is essential to meet the growing needs of this vulnerable population.

## Introduction

Population aging is an increasingly significant global phenomenon. According to United Nations projections, the proportion of people aged 65 and older is expected

**Data availability statement:** Full data set is available as Supporting information.

**Funding:** The study was supported by a grant from SIDACTION (Aide à la recherche 21-1-AEQ-12944). The grand recipient was BT. The funder had no role in study design, data collection and analysis, decision to publish, or preparation of the manuscript.

**Competing interests:** The authors have declared that no competing interests exist.

to rise from 1 in 11 in 2019–1 in 6 by 2050 [1]. Although this demographic transition is more advanced in high-income countries, it also affects low- and middle-income countries, where social and healthcare systems often struggle to adapt. In these settings, aging—long considered a marginal issue—is gradually emerging as a major public health concern.

In Cameroon, individuals aged 65 and above have accounted for approximately 2.7% of the national population in recent years (from 2.72% in 2020 to 2.77% in 2023), according to World Bank data [2]. Despite this relatively small proportion, older adults represent a high-risk group in terms of health, due to the high prevalence of chronic diseases, physical frailty, and socioeconomic vulnerability.

Traditionally, older adults in Cameroon held a respected social status, associated with wisdom, knowledge transmission, and intergenerational cohesion. The extended family played a central role in their care, sustained by strong intergenerational solidarity. However, social transformations, rapid urbanization, changing family structures, and economic hardship have deeply disrupted this model. Today, many elderly individuals live in isolation and must cope with their basic needs,—particularly healthcare,—on their own or with limited resources.

Cameroon's legal framework acknowledges the right of older adults to social protection. The Constitution states that "the Nation protects the elderly" and that "descendants have a duty to support their ascendants" [3]. Complementary legislation also addresses the protection of this population group [4]. In practice, however, existing mechanisms remain limited and largely ineffective. The formal social security system, based on International Labor Organization (ILO) principles, covers only about 10% of the population and fewer than 15% of workers, mostly from the formal sector [5]. Consequently, nearly 90% of the working population,—primarily informal workers,—is excluded [6].

Within this context, the absence of effective universal health coverage mechanisms exposes older adults to a high risk of forgoing or delaying care. Out-of-pocket payments remain the primary method of healthcare financing in Cameroon, further increasing their vulnerability. The management of chronic diseases,—common among older adults,—is particularly costly, both financially and socially. Despite the relevance and timeliness of the topic, few studies have been conducted on the direct financing of healthcare by the elderly in the Cameroonian context. Studies among this population focused more on their health conditions than on their financing sources. The city of Yaounde was selected because of the availability of different levels of healthcare services and its cosmopolitan character.

This study aims to assess the direct healthcare expenditures incurred by older adults during routine outpatient consultations in Yaounde, in the absence of universal health coverage. Additional objectives were to describe the pathological profile of the study population and to evaluate healthcare financing patterns.

## Methods

### Design and setting

This was a descriptive, cross-sectional, quantitative study conducted between February 8 and March18 2023. It was conducted in four health districts with high coverage

(> 400,000 inhabitants) in the city of Yaounde, Cameroon, within which four health facilities were selected: three referral hospitals,—the Military Hospital (HMY), the Central Hospital (HCY), and the University Teaching Hospital (CHUY),—and one intermediate-level facility, the Odza District Hospital. Referral hospitals were targeted because of their specialized care services, which are in high demand among the elderly (cardiology, rheumatology, geriatrics, endocrinology, etc.). The last healthcare facility was included to assess cost differences between central and intermediate-level institutions. This study was part of the "*Grand Age et VIH au Cameroun et au Sénégal: Anthropologie de la maladie et du vieillissement*" research project.

## Participants

The target population consisted of individuals aged 60 years and older who attended outpatient consultations at the selected institutions. All eligible patients were consecutively recruited after their medical consultations during the period of data collection. Eligible patients were individuals, both men and women, aged 60 and over, who had visited one of the four hospitals in the study for an outpatient consultation. Inclusion criteria were: (1) having been seen by a physician, (2) being able to provide informed consent, and (3) having completed the questionnaire in its entirety. Exclusion criteria were: (1) inability to communicate in French, (2) poor health preventing completion of the questionnaire, and (3) missing data on funding sources and healthcare expenditures related to the consultation. To assess the sample size, we used the formula:

$$n = \frac{z2\ pq}{d2}$$

Where q = 1-p; n = sample size; at 95% confidence interval, z = 1.96; we used a 5% marginal error (d)=0.05; 'p', the proportion of elderly person presenting with at least one health complain is estimated at 90% = 0.9. This gives a sample size n ≈ 140 [7]. A convenience sample of 170 participants was therefore recruited using a 10% refusal rate.

## Ethical considerations

The study received ethical approval from the National Ethics Committee of Cameroon. Administrative authorizations were obtained from all participating healthcare facilities. All participants gave free and written informed consent before inclusion in the study. Confidentiality of personal data was strictly maintained throughout the research, in accordance with ethical principles for health research.

## Data collection and procedures

Data were collected using a structured, closed-ended questionnaire administered through face-to-face interviews. The questionnaire consisted of five sections: (i) socio-demographic profile (7 questions); (ii) pathological profile (5 questions); (iii) direct health expenditures (6 questions); (iv) access to care; and (v) an open-ended question on social protection. Interviews were conducted after obtaining informed consent. For participants with medical prescriptions, information on medications and complementary tests was gathered using photographs or copies of prescriptions and test request forms.

## Data analysis

Data were entered and analysed using SPSS version 20.0. Continuous variables were described using means and standard deviations, while categorical variables were presented as frequencies and percentages. The variable of interest in the study was care financing modalities. Out-of-pocket expenditure was divided into two categories: (1) direct individual payments supported by the patient him/herself, and (2) family or household contributions. The variable *marital status* was categorised from four groups (single, divorced, married, widowed) into two: *single*, encompassing single and divorced participants, and *in a relationship* including both married and informal partners. A similar process was carried out with the

variable *professional status* (farmer, trader, employee, retired, unemployed), then analysed as *income status* with four groups: including those economically active either in the formal or the informal sectors, those retired and those with no income at all. These new categories were created in order to simplify data analysis and facilitate understanding of the results. Data were independent and categorical therefore associations between the variable of interest and demographic data (sex, age, marital status, employment status) were assessed using Chi-square tests, with a significance threshold set at $p < 0.05$. Medication costs were estimated using the Med Index application, and laboratory test costs were calculated based on the official price list of a reference laboratory (Pasteur Centre).

## Results

### Sociodemographic profile

A total of 170 older adults were included in the study, with a predominance of female participants (sex ratio: 0.81). The mean age was 68.4 (±1.2) years, with no significant difference between genders. The detailed socio-demographic characteristics of the study population are presented in Table 1.

Among participants, 31.7% were engaged in income-generating activities, primarily in trade (48.2%) and agriculture (25.9%), while nearly 30% had no income at all. The average number of children was 4.4 (±0.5). While 82.1% of participants reported having at least one financially independent child, only 37.3% stated that all their children were financially autonomous.

### Pathological profile

In 70.6% of cases, the consultation was a first-time visit. The reason for consultation were in 94.1% of participants related to age-associated health conditions. These are illustrated in Fig 1.

This situation contributed to frequent medical consultations and increased healthcare expenditure.

### Access and costs of care

Older adults' healthcare was primarily financed either through personal resources (43.5%) or family support (47.1%) representing a total out-of-pocket expenditure of 90.6%. The family support was predominantly provided by children (92.3%). Only 14 participants (8.2%) had health insurance coverage, with notable disparities based on marital status and

**Table 1. Sociodemographic characteristics.**

| Variables | Categories | n (%) |
|---|---|---|
| Sex | Female | 94 (55.3) |
| | Male | 76 (44.7) |
| Age (years) | [60-65] | 64 (37.6) |
| | [65-70] | 54 (31.8) |
| | [70-75] | 30 (17.6) |
| | [75-80] | 14 (8.3) |
| | [80-85] | 8 (4.7) |
| Marital status (N = 164) | Single | 62 (37.8) |
| | In a relationship | 102 (62.2) |
| Income | Informal sector | 40 (23.5) |
| | Formal sector | 14 (8.2) |
| | Retired | 66 (38.9) |
| | No income | 50 (29.4) |

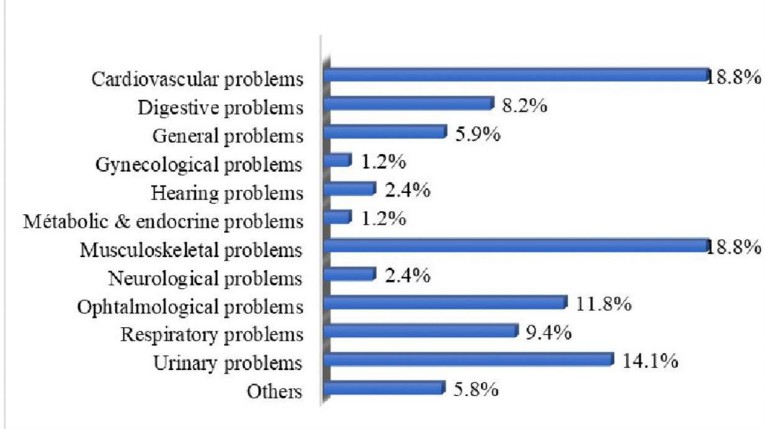

**Fig 1. Reasons for medical consultation.** Additionally, 84.5% of participants reported having at least one chronic comorbidity, the most common being hypertension, diabetes, and osteoarticular disorders (Fig 2).

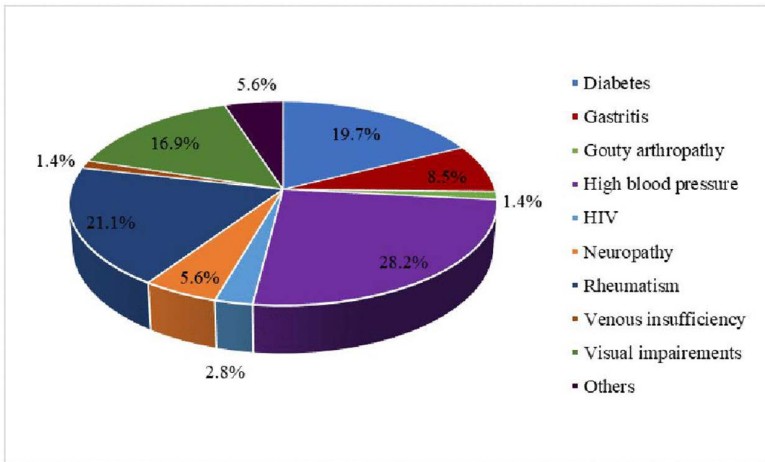

**Fig 2. Distribution of comorbidities (more than one answer possible).**

employment status (Table 2). Most participants (69.4%) attended medical consultations unaccompanied. When accompanied, they were most often supported by a child (61.5%), a spouse (23.1%), or a grandchild (15.4%). Most respondents were living in Yaounde (63.6%), while others travelled from surrounding towns (27.3%) or other regions of the country (9.1%). Public transportation was the most used means of travel (68.2%), with an average transport cost of 1,939 XAF (±735.9), or approximately €2.96.

Consultation fees varied according to the healthcare facility level, averaging 5,000 XAF (€7.62) at tertiary hospitals (HCY, HMY, CHUY) compared to 2,000 XAF (€3.05) at the primary-level Odza District Hospital. Following the consultation, 64.7% of patients received prescriptions that included medications, diagnostic tests, or specialist referrals. Medications constituted the main expense for 40% of patients, followed by medical tests (30.6%) and lifestyle-related recommendations (22.4%). The average cost of prescribed medications was 16,201 XAF (±5,909), approximately €24.70, while diagnostic tests cost an average of 24,818 XAF (±21,642), about €37.85. The total average cost of medical consultation for this older population—including consultation fees, transportation, and prescribed care—was therefore 42,958 XAF

**Table 2. Cross analysis of funds' sources and sociodemographic profile.**

| | Own funds | Family support | Insurance | p-value |
|---|---|---|---|---|
| **Sex** | | | | |
| Female | 30 | 58 | 6 | 0.021 |
| Male | 44 | 22 | 8 | |
| **Age (years)** | | | | |
| [60-65] | 22 | 36 | 4 | 0.430 |
| [65-70] | 28 | 20 | 6 | |
| [70-75] | 18 | 8 | 4 | |
| [75-80] | 6 | 8 | 0 | |
| [80-85] | 0 | 8 | 0 | |
| **Marital Status** | | | | |
| Single | 22 | 38 | 2 | 0.045 |
| In couple | 50 | 38 | 12 | |
| **Employment status** | | | | |
| Employed | 62 | 42 | 14 | 0.040 |
| Unemployed | 12 | 38 | 0 | |

(€65.49). Despite the presence of family solidarity mechanisms, 56.6% of older adults reported difficulty in covering all medical expenses. Consequently, 38.5% resorted to self-medication, and 15.4% used traditional medicine.

## Discussion

This study highlights the socio-demographic, health, and economic characteristics of older adults attending hospital consultations in Yaounde. The study population—predominantly female, relatively educated, and often accompanied—appears to represent a specific subgroup: urban older adults with a certain level of access to healthcare services. This limits the representativeness of the sample in comparison to the broader population of older adults in Cameroon, many of whom live in more vulnerable, isolated, or rural conditions. The observed female predominance is consistent with known patterns of higher longevity and greater morbidity among women, particularly regarding chronic illnesses and disability [8,9].

The data indicates diverse sources of income among participants. Among those who were working, few held positions in the formal sector, with the majority relying on informal activities. The high proportion of retired and participants with no income contrasts with national data showing that most older adults remain engaged in active income generation [10]. Furthermore, engagement in the informal sector highlights the precarious and flexible nature of livelihood strategies, which may influence participants' ability to finance routine healthcare. Similar findings have been reported in Burkina Faso [11], raising concerns about equity in healthcare access under conditions of widespread economic insecurity.

Marital status also appeared to play a protective role: older adults living with a partner benefited from better financial and logistical support (e.g., insurance coverage, cost-sharing, transportation, and healthcare decision-making). However, this protection may be fragile if the partner is also elderly, ill, or financially dependent.

Our findings reveal nuanced patterns of healthcare financing among older adults in Cameroon. While only one-third of participants reported that all their children were financially independent, family support remains a major source of health care funding, covering, primarily through children's contributions. On one hand, intergenerational solidarity persists: adult children continue to assist their parents, reflecting enduring moral and emotional bonds. On the other hand, this solidarity operates under economic constraints. Many younger adults face unemployment or underemployment [12], limiting their capacity to provide substantial financial support. Similar patterns have been observed across sub-Saharan Africa, where

intergenerational support remains central but is increasingly mediated by economic vulnerability and labor-market instability among younger generations [13]. Policy interventions aimed at strengthening financial protection for older adults should account for the limits of family-based support in contexts of generalized economic vulnerability. From a medical standpoint, all participants presented with at least one chronic comorbidity, primarily musculoskeletal, cardiovascular, ophthalmologic, or metabolic disorders. These findings align with studies in Morocco [14], where rheumatologic diseases accounted for 33.3% of reported conditions, followed by ophthalmologic (14.6%) and cardiovascular (9.6%) conditions. Such multimorbidity increases the need for health services (consultations, prescriptions, tests) and significantly raises healthcare costs. A previous study in Yaounde showed that individuals with chronic conditions used more services and spent more on healthcare: in 2018, a diabetic patient had up to 97 times more outpatient visits and incurred 83 times higher medication costs than a non-diabetic individual [15]. These results underscore the urgent need for comprehensive and subsidized chronic disease management programs to alleviate financial pressure on this vulnerable group.

The fact that most participants were consulting for the first time suggests gaps in prior healthcare access, highlighting the importance of developing innovative care delivery strategies. Nearly all costs were borne by households, in a country where only 2.06% of older adults are covered by health insurance [16]. Retirees from the formal sector had better coverage but remain a minority in an economy that is 80% informal [17]. In the absence of effective social protection mechanisms, older adults must assume catastrophic health expenditures on their own—especially women, widows, and those whose children are not financially autonomous [18]. These groups, among the most vulnerable, depend almost entirely on intergenerational solidarity.

The financial burden is exacerbated by high direct medical costs. The average healthcare expenses reported in our study (transportation, tests, medications)—sometimes exceeding half of the national minimum monthly wage [19]—indicate that healthcare remains largely unaffordable for the poorest segments of the population. This financial strain severely restricts access to care for low-income older adults and challenges the principle of universality embedded in the Sustainable Development Goals. As in Uganda, where 45% of older adults spend more than half of their income on health [20], medications and diagnostic tests represented the largest share of total costs. These findings may explain the frequent treatment discontinuation, care avoidance, and reliance on self-medication—practices that compromise treatment outcomes.

Self-medication—often combined with traditional remedies [21]—emerged as a coping strategy in the face of financial hardship but carries serious health risks, including drug interactions, delayed diagnoses, poisoning, and clinical complications. These practices highlight the urgent need to strengthen therapeutic education, preventive care, and safe access to essential medicines for older adults.

Given the findings of this study, several structural reforms are urgently needed to ensure equitable access to healthcare for older adults, whose vulnerability is increasing within the current socio-economic and demographic context. First, the implementation of a subsidized geriatric care package could significantly improve access by covering routine consultations, diagnostic tests, and essential medications. This package should include capped co-payments or full exemptions for older adults without regular income [22]. Second, there is a pressing need to develop health insurance products tailored to the informal sector—designed to be modular, affordable, and solidarity-based. Mechanisms such as voluntary family contributions could help extend universal coverage to non-salaried workers and their dependents.

Additionally, the deployment of mobile geriatric clinics would reduce geographical and logistical barriers by delivering care directly to older adults living in densely populated urban neighborhoods and peri-urban areas. Strengthening therapeutic education programs is also essential to raise awareness about the risks associated with self-medication and to promote better self-management of chronic conditions. Finally, formal support for family caregivers—through mechanisms such as solidarity-based taxation or the creation of a dedicated social fund for non-salaried older adults—would help reinforce intergenerational solidarity and ease the economic burden on households. In the context of a rapid demographic transition, the coordinated adoption of these measures is essential not only to preserve the autonomy and dignity of older adults, but also to sustainably reduce health inequities in Cameroon's aging population.

## Strengths and limitations

This study has several strengths. First, it provides detailed, empirical data on healthcare expenditures and financing strategies among older adults in Yaounde, a population often underrepresented in health systems research in Cameroon. Second, the inclusion of multiple healthcare facilities, covering both tertiary and intermediate-level hospitals, enhances the diversity of the sample and allows for comparisons across different levels of care. Third, the use of structured face-to-face interviews, supplemented by photographic documentation of prescriptions and test requests, improves the accuracy and reliability of reported costs.

However, some limitations should be acknowledged. The cross-sectional design limits the ability to infer causal relationships between socio-demographic factors and healthcare financing strategies. The study was conducted in an urban setting, which may limit the generalizability of findings to rural areas, where access to care and family support structures may differ. Additionally, the relatively small sample size may reduce statistical power for subgroup analyses. Despite these limitations, the study provides valuable insights into the economic challenges faced by older adults in Cameroon and highlights critical areas for policy intervention.

## Conclusion

These findings underscored that older adults in Cameroon finance primarily their healthcare through out-of-pocket payments, a situation that severely limits access to care, especially among low-income households. The lack of risk-pooling mechanisms and the low rate of insurance coverage exposes this population to economically and medically precarious coping strategies, including delaying or forgoing care, incurring debt, self-medication, and reliance on traditional medicine. Addressing the dual challenge of rising healthcare needs and financial vulnerability requires structural reforms.

## Supporting information

**S1 Data. Complet Data Set.**
(XLSX)

**S1 Checklist. PlosOne Checklist for Human Subject Research.**
(DOCX)

## Author contributions

**Conceptualization:** Marie-José Essi, Gabriele Laborde-Balen, Bernard Taverne, Laura Ciaffi.

**Data curation:** Johanne A. Abossolo, Divine H. Aba'a.

**Formal analysis:** Johanne A. Abossolo.

**Funding acquisition:** Bernard Taverne, Laura Ciaffi.

**Methodology:** Marie-José Essi.

**Supervision:** Marie-José Essi, Gabriele Laborde-Balen, Laura Ciaffi.

**Validation:** Marie-José Essi, Bernard Taverne, Laura Ciaffi.

**Writing – original draft:** Marie-José Essi, Johanne A. Abossolo, Divine H. Aba'a.

**Writing – review & editing:** Marie-José Essi, Sidonie L. Ndjebet, Gabriele Laborde-Balen, Bernard Taverne, Laura Ciaffi.

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
