## [Decision Letter · Decision Letter 0]

16 Oct 2025

PGPH-D-25-02145

Assessment of healthcare financing by older adults during routine consultations in Cameroon

Dear Dr. ciaffi,

Thank you for submitting your manuscript to PLOS Global Public Health. After careful consideration, we feel that it has merit but does not fully meet PLOS Global Public Health’s publication criteria as it currently stands.

Therefore, we invite you to submit a revised version of the manuscript that addresses the points raised during the review process.

We look forward to receiving your revised manuscript.

Kind regards,

Anil Gumber, Ph.D.

Academic Editor

Journal Requirements:

Additional Editor Comments (if provided):

Thank you for submitting an interesting paper. However, the reviewers have raised serious issues about the analysis and quality of the statistical tools used. I hope you will be able to revise the paper.

Reviewers' comments:

Reviewer's Responses to Questions

**Comments to the Author**

1. Does this manuscript meet PLOS Global Public Health’s publication criteria? Is the manuscript technically sound, and do the data support the conclusions? The manuscript must describe methodologically and ethically rigorous research with conclusions that are appropriately drawn based on the data presented.? Is the manuscript technically sound, and do the data support the conclusions? The manuscript must describe methodologically and ethically rigorous research with conclusions that are appropriately drawn based on the data presented.

Reviewer #1: Yes

Reviewer #2: Partly

Reviewer #3: Partly

2. Has the statistical analysis been performed appropriately and rigorously?

Reviewer #1: No

Reviewer #2: Yes

Reviewer #3: Yes

3. Have the authors made all data underlying the findings in their manuscript fully available (please refer to the Data Availability Statement at the start of the manuscript PDF file)?

The PLOS Data policy requires authors to make all data underlying the findings described in their manuscript fully available without restriction, with rare exception. The data should be provided as part of the manuscript or its supporting information, or deposited to a public repository. For example, in addition to summary statistics, the data points behind means, medians and variance measures should be available. If there are restrictions on publicly sharing data—e.g. participant privacy or use of data from a third party—those must be specified.requires authors to make all data underlying the findings described in their manuscript fully available without restriction, with rare exception. The data should be provided as part of the manuscript or its supporting information, or deposited to a public repository. For example, in addition to summary statistics, the data points behind means, medians and variance measures should be available. If there are restrictions on publicly sharing data—e.g. participant privacy or use of data from a third party—those must be specified.

Reviewer #1: Yes

Reviewer #2: Yes

Reviewer #3: Yes

4. Is the manuscript presented in an intelligible fashion and written in standard English?

Reviewer #1: Yes

Reviewer #2: No

Reviewer #3: Yes

Reviewer #1: Interesting read.

Pertinent topic and valuable data captured.

Please consider remarks and questions below:

Your study shows a 43.5% out of pocket expenditure. This is relatively very low compared to multiple other reports for Cameroon that put out of pocket expenditure above 70%. Unless a serious selection bias is to be considered, kindly state (or review) your definition of out of pocket health expenditure. Traditionally out of pocket expenditures accounts for payment outside of an insurance scheme.

Confer definitions here: (https://databank.worldbank.org/metadataglossary/health-nutrition-and-population-statistics/series/SH.UHC.OOPC.25.TO).

According to this, household spending including from family members which was not part of a health fund or insurance scheme of any kind is also out of pocket expenditure. In your paper, contributions for family was not considered as out of pocket.

"The high proportion of retirees and inactive individuals contrasts with national data showing that most older adults remain economically active".

Your statement above contrasts with your finding that 70.5% of participants had an income. Can you clarify this reference to a high proportion of retirees in your study population? Clear definitions of income categories will help.

"The study further reveals signs of weakened family solidarity: only one-third of participants reported that all their children were financially independent, in a context marked by high youth unemployment. In such cases, the family traditionally considered a safety net—may in fact contribute to economic precarity".

Your statement above contrasts with your finding that health care spending for your participants was covered by families 47.1% (over 90% by children) only second to self sponsored care. Can you please explain? In addition, family solidarity will not be defined by financial independence of the children. Unless you clarify and give context to this with comparative figures, this conclusion might be inappropriate.

Reviewer #2: The authors have conducted univariate and bivariate analyses, but no multivariate analysis. There is also a need for improving the language of writing to academic requirements. The authors are also suggested to use STROBE checklist for cross-sectional studies for adherence to reporting standards.

Reviewer #3: Great work overall.

1) Introduction- address the need for the study. Any similar study in the past? Why was Yaounde chosen?

2) Table 1- Format properly. Remove the commas and insert decimal points for percentages.

3) Table 2 – Also needs the same correction.

4) Table 1- Age column totals up to 99.9%.

5) Figure 1- Total percentage adds up to 131.3%.

6) Figure 2- Total percentage adds up to 109.8%.

7) Discussion- Please mention the strengths and limitations in a paragraph exclusively.

8) Conclusion- Please limit the word count within 200 words.

Thank You

**Do you want your identity to be public for this peer review?** For information about this choice, including consent withdrawal, please see our Privacy Policy..

Reviewer #1: No

Reviewer #2: **Yes:** Denny JohnDenny JohnDenny JohnDenny John

Reviewer #3: **Yes:** DR. DENNY MATHEW JOHNDR. DENNY MATHEW JOHNDR. DENNY MATHEW JOHNDR. DENNY MATHEW JOHN

Department of Community Medicine

Kerala Medical College Hospital

Palakkad, Kerala, India

---

## [Decision Letter · Decision Letter 1]

12 Feb 2026

PGPH-D-25-02145R1

Assessment of healthcare financing by older adults during routine consultations in Cameroon

Dear Dr. ciaffi,

Thank you for submitting your manuscript to PLOS Global Public Health. After careful consideration, we feel that it has merit but does not fully meet PLOS Global Public Health’s publication criteria as it currently stands. Therefore, we invite you to submit a revised version of the manuscript that addresses the points raised during the review process.

The manuscript has been evaluated by two reviewers, and their comments are available below.

The reviewers have raised a few remaining concerns. Could you please revise the manuscript to carefully address the concerns raised?

We look forward to receiving your revised manuscript.

Kind regards,

Johanna Pruller, Ph.D.

PLOS Staff Editor

Journal Requirements:

Additional Editor Comments (if provided):

Reviewers' comments:

Reviewer's Responses to Questions

**Comments to the Author**

Reviewer #2: All comments have been addressed

Reviewer #3: (No Response)

publication criteria? Is the manuscript technically sound, and do the data support the conclusions? The manuscript must describe methodologically and ethically rigorous research with conclusions that are appropriately drawn based on the data presented.? Is the manuscript technically sound, and do the data support the conclusions? The manuscript must describe methodologically and ethically rigorous research with conclusions that are appropriately drawn based on the data presented.

Reviewer #2: Yes

Reviewer #3: Yes

3. Has the statistical analysis been performed appropriately and rigorously?

Reviewer #2: Yes

Reviewer #3: Yes

4. Have the authors made all data underlying the findings in their manuscript fully available (please refer to the Data Availability Statement at the start of the manuscript PDF file)?

The PLOS Data policy requires authors to make all data underlying the findings described in their manuscript fully available without restriction, with rare exception. The data should be provided as part of the manuscript or its supporting information, or deposited to a public repository. For example, in addition to summary statistics, the data points behind means, medians and variance measures should be available. If there are restrictions on publicly sharing data—e.g. participant privacy or use of data from a third party—those must be specified.requires authors to make all data underlying the findings described in their manuscript fully available without restriction, with rare exception. The data should be provided as part of the manuscript or its supporting information, or deposited to a public repository. For example, in addition to summary statistics, the data points behind means, medians and variance measures should be available. If there are restrictions on publicly sharing data—e.g. participant privacy or use of data from a third party—those must be specified.

Reviewer #2: Yes

Reviewer #3: Yes

5. Is the manuscript presented in an intelligible fashion and written in standard English?

Reviewer #2: Yes

Reviewer #3: Yes

Reviewer #2: The manuscript is acceptable in the current form.

Reviewer #3: Great work authors. All concerns have been addressed except the pathological profile (figures 1 & 2). Are those mutually exclusive events? The percentages are not totaling to 100.

**Do you want your identity to be public for this peer review?** For information about this choice, including consent withdrawal, please see our Privacy Policy..

Reviewer #2: **Yes:** Denny JohnDenny JohnDenny JohnDenny John

Reviewer #3: **Yes:** DR. DENNY MATHEW JOHNDR. DENNY MATHEW JOHNDR. DENNY MATHEW JOHNDR. DENNY MATHEW JOHN

Kerala Medical College Hospital, Palakkad

India

---

## [Decision Letter · Decision Letter 2]

12 Mar 2026

Assessment of healthcare financing by older adults during routine consultations in Cameroon

PGPH-D-25-02145R2

Dear Dr ciaffi,

We are pleased to inform you that your manuscript 'Assessment of healthcare financing by older adults during routine consultations in Cameroon' has been provisionally accepted for publication in PLOS Global Public Health.

Best regards,

Julia Robinson

Executive Editor

Reviewer Comments (if any, and for reference):

Reviewer's Responses to Questions

**Comments to the Author**

Reviewer #3: All comments have been addressed

publication criteria? Is the manuscript technically sound, and do the data support the conclusions? The manuscript must describe methodologically and ethically rigorous research with conclusions that are appropriately drawn based on the data presented.? Is the manuscript technically sound, and do the data support the conclusions? The manuscript must describe methodologically and ethically rigorous research with conclusions that are appropriately drawn based on the data presented.

Reviewer #3: (No Response)

3. Has the statistical analysis been performed appropriately and rigorously?

Reviewer #3: (No Response)

4. Have the authors made all data underlying the findings in their manuscript fully available (please refer to the Data Availability Statement at the start of the manuscript PDF file)?

The PLOS Data policy requires authors to make all data underlying the findings described in their manuscript fully available without restriction, with rare exception. The data should be provided as part of the manuscript or its supporting information, or deposited to a public repository. For example, in addition to summary statistics, the data points behind means, medians and variance measures should be available. If there are restrictions on publicly sharing data—e.g. participant privacy or use of data from a third party—those must be specified.requires authors to make all data underlying the findings described in their manuscript fully available without restriction, with rare exception. The data should be provided as part of the manuscript or its supporting information, or deposited to a public repository. For example, in addition to summary statistics, the data points behind means, medians and variance measures should be available. If there are restrictions on publicly sharing data—e.g. participant privacy or use of data from a third party—those must be specified.

Reviewer #3: (No Response)

5. Is the manuscript presented in an intelligible fashion and written in standard English?

Reviewer #3: (No Response)

Reviewer #3: (No Response)

**Do you want your identity to be public for this peer review?** For information about this choice, including consent withdrawal, please see our Privacy Policy..

Reviewer #3: **Yes:** DR. DENNY MATHEW JOHNDR. DENNY MATHEW JOHNDR. DENNY MATHEW JOHNDR. DENNY MATHEW JOHN

Department of Community Medicine

Kerala Medical College Hospital

Palakkad, Kerala, India
